# Integration of Metabolomic and Transcriptomic Analyses Reveals the Molecular Mechanisms of Flower Color Formation in *Prunus mume*

**DOI:** 10.3390/plants13081077

**Published:** 2024-04-11

**Authors:** Ruyi Wang, Xin Yang, Tao Wang, Baohui Li, Ping Li, Qin Zhang

**Affiliations:** 1College of Landscape and Tourism, Hebei Agricultural University, Baoding 071000, China; ruyi02061005@163.com (R.W.); 15232294478@139.com (X.Y.); 2College of Forestry, Hebei Agricultural University, Baoding 071000, China; 3China National Botanical Garden, Beijing 100089, China

**Keywords:** *Prunus mume*, flower color, anthocyanin synthesis pathway, gene regulatory network

## Abstract

Flower color is an important trait that affects the economic value of *Prunus mume*, a famous ornamental plant in the Rosaceae family. *P. mume* with purple–red flowers is uniquely charming and highly favored in landscape applications. However, little is known about its flower coloring mechanism, which stands as a critical obstacle on the path to innovative breeding for *P. mume* flower color. In this study, transcriptomic and targeted metabolomic analyses of purple–red *P. mume* and white *P. mume* were performed to elucidate the mechanism of flower color formation. In addition, the expression patterns of key genes were analyzed using an RT-qPCR experiment. The results showed that the differential metabolites were significantly enriched in the flavonoid synthesis pathway. A total of 14 anthocyanins emerged as the pivotal metabolites responsible for the differences in flower color between the two *P. mume* cultivars, comprising seven cyanidin derivatives, five pelargonium derivatives, and two paeoniflorin derivatives. Moreover, the results clarified that the metabolic pathway determining flower color in purple–red *P. mume* encompasses two distinct branches: cyanidin and pelargonidin, excluding the delphinidin branch. Additionally, through the integrated analysis of transcriptomic and metabolomic data, we identified 18 key genes responsible for anthocyanin regulation, thereby constructing the gene regulatory network for *P. mume* anthocyanin synthesis. Among them, ten genes (*PmCHI*, *PmGT2*, *PmGT5*, *PmGST3*, *PmMYB17*, *PmMYB22*, *PmMYB23*, *PmbHLH4*, *PmbHLH10*, and *PmbHLH20*) related to anthocyanin synthesis were significantly positively correlated with anthocyanin contents, indicating that they may be the key contributors to anthocyanin accumulation. Our investigation contributes a novel perspective to understanding the mechanisms responsible for flower color formation in *P. mume*. The findings of this study introduce novel strategies for molecular design breeding aimed at manipulating flower color in *P. mume*.

## 1. Introduction

Plant flower color is a significant biological trait, as it attracts pollinators, facilitates reproduction, and enhances resistance to various environmental stresses. Furthermore, flower color serves as a pivotal ornamental feature that directly affects the aesthetic and economic value of garden plants. Flavonoids are the major pigments in most plant flowers, fruits, and seeds [1]. They encompass seven categories: chalcones, flavonoids, flavonols, flavandiols, proanthocyanidins, aurones, and anthocyanins [2]. Anthocyanins, a subset of flavonoids, play a central role in flower color formation. Anthocyanins have six different forms: cyanidin, peonidin, delphinidin, petunidin, malvidin, and pelargonidin; these forms contribute to a range of colors, including red, orange, blue, and purple [3]. Six key enzymes in the flavonoid synthesis pathway play key roles in anthocyanin synthesis, including chalcone synthase (CHS), chalcone isomerase (CHI), dihydroflavonol reductase (DFR), flavanone-3-hydroxylase (F3H), anthocyanidin synthase (ANS), and UDP glucose flavonoid 3-glucosyltransferase (UFGT). The MYB, bHLH, and WD40 transcription factor families often form MBW transcription complexes to regulate anthocyanin synthesis. In recent years, the WRKY transcription factor family has also been shown to regulate anthocyanin synthesis [4,5]. In *Centaurea cyanus*, transcription factor genes such as *CcMYB6*-1 and *CcbHLH1* collaboratively regulate cyanidin biosynthesis, while gene mutations in *CcbHLH1* and *CcF3′H* in distinct varieties affect anthocyanin synthesis [6]. In *Dianthus caryophyllus*, bHLH and MYB jointly regulate anthocyanin synthesis, and WRKY44 interacts with *DsDFR* to influence petal edge coloring [7].

*Prunus mume* Sieb. et Zucc, also known as Mei Flower, is an important ornamental plant in the *Prunus* genus of the Rosaceae family. It has a cultivation history of more than 3000 years in China. Due to its flowering in early spring, strong cold and disease resistance, and tolerance of barren soil, *P. mume* symbolizes a strong, noble, and modest character in Chinese traditional culture. This species has high ornamental and economic value and is not only used as an ornamental plant in gardens and courtyards but is also cultivated as a bonsai, cut flower, or medicinal plant. Notably, this was the first *Prunus* species to undergo complete whole-genome sequencing [8]. In 2018, genomic data were collected by resequencing 348 *P. mume* accessions [9]. These data laid the foundation for genetic investigations into *P. mume* floral color. Previous studies on flower colors of *P. mume* have mostly focused on phenotype and anthocyanin contents [10,11], while the specific anthocyanin synthesis pathway in *P. mume* flowers has not been elucidated [12]. The red and white flower series of *P. mume* is predominant among its cultivated varieties, enjoying widespread popularity and application. Yet the molecular mechanisms governing the color differences in *P. mume* flowers remain unclear, impeding the advancement of molecular design breeding techniques for flower color enhancement.

Metabolomic analysis allows for the comprehensive identification of metabolites associated with flower color synthesis, bridging the gap between phenotype and gene expression. The application of high-throughput sequencing technology facilitates the examination of transcriptomes, thereby simplifying the identification of differentially expressed genes (DEGs) in various samples and the measurement of target gene expression levels [13]. These techniques can be combined to predict gene function, construct essential regulatory networks, and identify pivotal regulatory genes. This dual approach opens new avenues for exploring the mechanisms governing flower color regulation, with successful applications in horticultural crop studies. For instance, the key genes involved in the regulation of anthocyanin synthesis in *Camellia reticulata* were identified upon analyzing the transcriptome and metabolome of white and pink flowers [14]. In addition, the mechanism of regulating sunflower anthocyanin synthesis via the *HaMYB1* gene has been elucidated through transcriptome and metabolome association analysis, which provides a new understanding of sunflower color formation [15].

In this study, extending previous studies on *P. mume* flower color, we applied metabolomics technology to unravel the detailed metabolic pathways linked to anthocyanin synthesis by identifying the flavonoid intermediate metabolite components and the differential metabolites (DAMs) of different flower colors. Simultaneously, transcriptome sequencing was performed to analyze the DEGs related to anthocyanin biosynthesis of different *P. mume* flower colors. Subsequently, key regulatory genes were identified via metabolome–transcriptome data correlation analysis, culminating in the construction of a gene regulatory network for *P. mume* anthocyanin synthesis. These findings lay a foundation for further investigations into the mechanisms underlying *P. mume* flower color synthesis and for proposing molecular breeding design strategies aimed at improving *P. mume* flower coloration.

## 2. Results

### 2.1. Identification of Flavonoids and Differentially Accumulated Metabolites between Purple–Red and White Flowers in P. mume

The PCA diagram of the two *P. mume* varieties indicated that the first main component accounted for 57.62% of the variation, whereas the second principal component accounted for 14.05%. This plot indicated considerable dissimilarity between the two *P. mume* samples, with samples within each group clustering together, indicating good repeatability (Appendix A).

A total of 579 metabolites were identified in two *P. mume* cultivars, encompassing 545 flavonoids and 34 tannins. Their content in purple–red flowers was significantly higher than that in white flowers (Appendix A, Appendix A). Secondary metabolites were classified into 29 chalcones, 3 aurones, 185 flavones, 155 flavonols, 14 anthocyanins, 37 flavanones, 16 flavanonols, 31 flavanols, 48 other flavonoids, 27 isoflavones, 13 proanthocyanidins, and 21 tannins. Among them, the content of flavonols was the highest, which was 2.58 × 10^9^ in white flower varieties, accounting for 60% of all flavonoid metabolites, and reached 3.02 × 10^9^ in purple–red flower varieties, accounting for 50%. The second was flavones; the content of flavones in white flower varieties was 2.58 × 10^8^, accounting for 21%, and the content in purple–red flower varieties was 1.48 × 10^9^, accounting for 24%. The most different flavonoid metabolite in the petals of the two varieties of *P. mume* was anthocyanins. Its content in white flower varieties was 3.50 × 10^7^, accounting for only 1%. In purple–red flower varieties, it reached 6.09 × 10^8^, accounting for 10% (Figure 1B).

A total of 245 DAMs were identified in the two cultivars’ flowers (fold change (FC) ≥ 2 or FC ≤ 0.5, probability (*p*-value) < 0.05, and VIP > 1), consisting of 136 up-regulated and 109 down-regulated metabolites. The KEGG enrichment analysis of these DAMs revealed enrichment in four pathways, exhibiting notable enhancement in the processes of flavonoid biosynthesis, anthocyanin biosynthesis, flavone and flavonol biosynthesis pathways, and isoflavonoid biosynthesis (Figure 1C). Anthocyanins emerged as the pivotal metabolites responsible for the differences in flower color between the two *P. mume* cultivars, consisting of fourteen identified anthocyanins, comprising seven cyanidin derivatives, five pelargonium derivatives, and two paeoniflorin derivatives. Interestingly, for all 13 anthocyanins, except pelargonidin-3-O-(6″-O-malonyl) glucoside, the amounts of those substances were notably elevated in purple–red flowers (Figure 1D). Notably, among the fourteen anthocyanins identified, the FC in cyanidin-3-O-(2″-O-glucosyl) rutinoside, cyanidin-3-O-gentiobioside, peonidin-3-O-arabinoside, pelargonidin-3-O-(6″-O-acetyl) glucoside, and pelargonidin-3-O-rutinoside were greater than six, and the accumulation in purple–red flowers was significantly up-regulated. Therefore, it was speculated that these five anthocyanins were the major contributors to the differences in flower color. This result also suggests that there are two branches of anthocyanin metabolism pathways, namely cyanidin and geranium, in the purple–red flowers of *P. mume* rather than the delphinidin branch.

### 2.2. Transcriptome Sequencing and Analysis of DEGs

To investigate the transcriptional regulation of the accumulation of flavonoid metabolites, transcriptome sequencing and assembly were performed on six samples (YD1, YD2, YD3, WYY1, WYY2, and WYY3) from both varieties. The samples were matched to the reference genome of *P. mume*. Following quality control and raw data filtering, a cumulative amount of 47.54 Gb of high-quality clean data was acquired, and each sample produced around 6 Gb of clean data. The Q30 base percentage of the samples exceeded 92.86%, and the GC content ranged between 43% and 48% (Appendix A). The alignment rate of clean reads to the reference genome in YD varied between 90.78% and 91.27%, while in WYY, it ranged from 88.85% to 90.39% (Appendix A). Utilizing these comparison results and gene positional information within the reference genome, we tabulated the read counts for each gene, normalized the mapped read counts, and considered the transcript length in the samples. The density map could show the trend of gene abundance change with the expression level in the sample and could reflect the interval of gene expression concentration in the sample. The gene expression density distributions of the samples are shown in Appendix A. In the comparison of the two *P. mume* varieties, 4658 DEGs were discovered (log_2_ fold change (|log_2_(FC)|) ≥ 1 and false discovery rate (FDR) < 0.05), with 2404 up-regulated and 2254 down-regulated (Appendix A). This result indicated that significant differences in gene expression levels were observed between the two *P. mume* cultivars, with the potential identification of crucial genes (Figure 2A and Appendix A).

### 2.3. KEGG Pathway Enrichment Analysis and Identification of DEGs Associated with Anthocyanin Synthesis

We performed GO secondary classification and enrichment analyses among distinct subgroups. The results from the GO-enriched directed acyclic graph revealed that DEGs were predominantly enriched for RNA modification (GO: 0009451) within the biological process category. Significant enrichment was seen in major branches of the molecular function category, including transferase activity, transferring one-carbon groups (GO: 0016741), methyltransferase activity (GO: 0008168), and RNA methyltransferase activity (GO: 0008173) (Appendix A). To further decipher the functions of DEGs within specific metabolic pathways, they were analyzed for KEGG pathway classification and enrichment. A total of 3352 Unigene sequences were enriched across 138 KEGG pathways. The pathways that showed the highest enrichment were metabolic pathway (Ko01100), biosynthesis of secondary metabolites (Ko01110), plant hormone signal transduction (Ko04075), ribosome biogenesis in eukaryotes (Ko03008), and galactose metabolism (Ko00052). The metabolic pathways contained the highest number of annotated genes. Additionally, we delved into secondary pathways, particularly those closely related to flower color, including phenylpropanoid biosynthesis (Ko00940), flavonoid biosynthesis (Ko00941), anthocyanin biosynthesis (Ko00942), isoflavonoid biosynthesis (Ko00943), flavone and flavonol biosynthesis pathway (Ko00944), and glutathione metabolism (Ko00480) (Figure 2B,C and Appendix A). We identified 13 genes responsible for anthocyanin synthesis, including 1 *CHI*, 1 *FLS*, 1 *DFR*, 1 *ANS*, 2 *ANR*, 1 *LAR*, 3 *UFGT*, and 3 *GST*. The CDS sequences and cloning primers of the above 13 genes are shown in Appendix A. The expression patterns of these genes in the two *P. mume* cultivars are depicted in Figure 2D. Among them, *PmCHI*, *PmANS*, *PmGT1*, *PmGT2*, *PmGT5*, *PmGST1*, *PmGST2*, and *PmGST3* were up-regulated in purple–red flowers, whereas *PmDFR*, *PmFLS*, *PmLAR*, *PmANR1*, and *PmANR2* were down-regulated in purple–red flowers.

### 2.4. RT-qPCR Validation of Genes Related to Anthocyanin Synthesis

With reference to the gene expression level in white flower variety YD, 11 candidate genes *(PmCHI*, *PmFLS*, *PmANR1*, *PmLAR*, *PmANS*, *PmGT5*, *PmGST2*, *PmGST3*, *PmMYB23*, *PmbHLH4*, *and PmMYB22*) related to anthocyanin synthesis were identified by RT-qPCR (Figure 3 and Appendix A). The results indicated that the expression levels of *PmCHI*, *PmANS*, *PmGT5*, *PmGST2*, *PmGST3*, and *PmMYB23* were up-regulated in purple–red flowers, while *PmFLS*, *PmANR1*, and *PmLAR* were down-regulated. In particular, the expression level of *PmGST3* in purple–red flowers reached eight times higher than in white flowers. In contrast, the expression level of *PmANR1* in purple–red flowers decreased by 2.6 times. The results demonstrated that the expression patterns of these 11 genes in YD and WYY aligned with transcriptome sequencing data.

### 2.5. Transcription Factor Analysis

Previous studies had indicated that numerous transcription factors played crucial regulatory roles in the synthesis of anthocyanins, among which the most reported were MYB, bHLH, WD40, and WRKY [4,5]. In this study, we investigated the relationship between MYB, bHLH, and WRKY transcription factors and the key genes associated with anthocyanin synthesis. Totals of 34 MYB, 30 bHLH, and 30 WRKY proteins were identified. The sequence IDs of some key genes are shown in Appendix A. Transcription factors and structural genes associated with anthocyanin synthesis with correlation coefficients greater than 0.9 and *p*-values less than 0.01 were chosen to create a correlation network diagram (Figure 4, Appendix A). The analysis revealed that 20 MYB, 16 bHLH, and 12 WRKY transcription factors were significantly correlated with genes related to anthocyanin synthesis. It could be seen from Figure 4 that each gene was significantly correlated with multiple transcription factors. We speculated that the structural genes of the anthocyanin synthesis pathway in *P. mume* were jointly regulated by multiple transcription factors.

### 2.6. Construction of Regulatory Networks Associated with Anthocyanin Synthesis

To clarify regulatory networks associated with anthocyanin synthesis, the correlation between 13 key genes of anthocyanin and anthocyanin content that were found to have high contents in purple–red flowers was analyzed (Figure 5, Appendix A). The results revealed significant correlations between 12 anthocyanins and five structural genes in the anthocyanin synthesis pathway (*p* < 0.01). Among them, *PmCHI*, *PmGT2*, *PmGT5*, and *PmGST3* were positively correlated with the content of anthocyanin in *P. mume*, while *PmANR1* was negatively correlated. Additionally, six MYBs, six bHLHs, and one WRKY transcription factor genes were significantly correlated with twelve anthocyanins. Among them, *PmMYB17*, *PmMYB22*, *PmMYB23*, *PmbHLH4*, *PmbHLH10*, and *PmbHLH20* were positively correlated with anthocyanin content, while *PmMYB5*, *PmMYB9*, *PmMYB27*, *PmbHLH2*, *PmbHLH12*, *PmbHLH16*, and *PmWRKY4* were negatively correlated with anthocyanin content. Accordingly, it could be inferred that these five structural genes and thirteen transcription factor genes might have regulatory functions in the process of anthocyanin production. Among them, the gene regulatory network consisting of ten genes (*PmCHI*, *PmGT2*, *PmGT5*, *PmGST3*, *PmMYB17*, *PmMYB22*, *PmMYB23*, *PmbHLH4*, *PmbHLH10*, and *PmbHLH20*) might play a critical role in promoting the accumulation of anthocyanosides.

To gain insight into the anthocyanin synthesis process in *P. mume*, we constructed a detailed pathway map of anthocyanin biosynthesis. This map provided a comprehensive visual representation of the biochemical steps, key metabolite contents, and gene expression during anthocyanin synthesis in *P. mume* (Figure 6). Consequently, the heightened expression of *CHI* in purple–red flowers might have led to an increased naringenin content. Conversely, the down-regulation of *FLS* expression in purple–red flowers diverted more dihydrokaempferol toward the flavonoid branch. The expression of *DFR* was down-regulated in purple–red flowers. Therefore, we speculated that *DFR* may be related to other metabolic processes in *P. mume*, and the key genes leading to the difference in flower color were downstream of *DFR*. The up-regulation of *LAR* and *ANR* expression in white flowers directed leucocyanidin toward the proanthocyanidin synthesis pathway, leading to notably higher levels of catechins and epicatechins. In contrast, the up-regulation of *ANS* expression in purple–red flowers stimulated the conversion of leucocyanidin and leucopelargonidin into cyanidin and pelargonidin in purple–red flowers while concurrently causing a reduction in catechins into cyanidin. Consequently, this yielded higher quantities of cyanidin-3-O-glucoside, pelargonidin-3-O-glucoside, pelargonidin-3-O-rutinoside, and peonidin-3-O-rutinoside in purple–red flowers, resulting in a distinct floral color disparity between the two varieties. Moreover, the *UFGT* and *GST* genes were up-regulated in purple–red flowers, serving as crucial contributors to the augmented accumulation of anthocyanins. In light of the aforementioned analysis, this study identified *CHI*, *ANR*, *UFGT*, and *GST* as potential candidate genes responsible for the different flower colors observed in YD and WYY.

## 3. Discussion

The identification of DAMs through metabolomics is widely used to elucidate the metabolic pathways of flower, leaf, and fruit colors in various plants. For instance, the identification of differential metabolites is employed to investigate the factors underlying the coloration of three distinct wheat grains, identifying cyanidin and peonidin as the principal anthocyanins responsible for color variation [16]. In the context of brown cotton fibers, most flavonoid-associated metabolites, such as myricetin, naringenin, catechins, epicatechin–epiafzelechin, and epigallocatechin, exhibit greater up-regulation than in white cotton fibers [17]. The composition and content of different flavonoid metabolites reveal the reasons for the flower color variation in *Rhododendron pulchrum* varieties [18]. In this study, we employed metabolomic approaches to systematically analyze the metabolic components in the petals of *P. mume*, unveiling the detailed branched pathways of anthocyanin metabolism in its flowers. The variation in flower color between two *P. mume* cultivars was primarily attributed to significant differences in anthocyanin content. Among these anthocyanins, cyanidin and pelargonidin displayed the most significant disparities, with a marked accumulation in purple–red flower varieties as opposed to white varieties, a trend mirrored in *Boehmeria nivea* leaf color [19]. The clarification of the anthocyanin metabolic pathway has paved the way for further investigation of the molecular regulatory mechanisms of flower coloration in *P. mume*.

Transcriptome analysis is an effective tool for studying color differences in plant flowers, fruits, and leaves. For instance, the key genes controlling fruit color were identified through transcriptome analysis of the white and red fruits of *Ailanthus altissima* [20]. Transcriptome analysis of *Acer palmatum*, featuring yellow and red leaves, provides new insights into the mechanisms underlying leaf color changes [21]. In a study of three *Lantana camara* varieties with distinct flower colors, analysis of DEGs revealed the molecular mechanisms responsible for color variation [22]. The key genes that cause color changes in tubers of *Pinellia ternata* were screened using transcriptome technology, providing a theoretical basis for studying purple skin formation [23]. The correlation analysis between differential metabolites and DEGs provides a more comprehensive explanation of plant color formation mechanisms. The combined analysis of transcriptome and metabolome for white-flowered *Sophora japonica* and red-flowered mutants indicates that cyanidin-3-O-glucoside and cyanidin-3-O-rutinoside are the key metabolites influencing its red color. Concurrently, the differential expressions of *F3′5′H*, *ANS*, *UFGT*, *bHLH*, and *WRKY* emerge as key genes, leading to their diverse colors [24]. Correlation analysis between metabolism and transcription at different developmental stages of purple and white flowers of *Dendrobium nobile* indicates that a reduction in anthocyanin content primarily accounted for the inability to form purple flowers [25]. Therefore, the identification of DEGs and correlation analysis of the transcriptional metabolism are helpful in understanding the molecular mechanisms underlying plant flower color formation, potentially offering genetic resources for flower color breeding.

Prior research has demonstrated the significance of MYB, bHLH, WD40, and WRKY as crucial transcription factors in the control of flavonoid synthesis in numerous plant species. These transcription factors can either autonomously or collaboratively regulate flower pigment synthesis. In *A. thaliana*, 339 MYB transcription factors were identified, with subfamilies S4, S5, S6, and S7 implicated in anthocyanin synthesis. The bHLH proteins that regulate anthocyanin production include TT8, EGL3, and MYC1 [26,27]. Additionally, the WD40 protein often forms transcriptional complexes with MYB and bHLH to regulate anthocyanin synthesis. Notably, WRKY transcription factors have been found to regulate anthocyanin production in *Freesia hybrida*, apples, and pears [28,29,30,31]. In *P. mume*, gene families encompassing R2R3-MYB, bHLH, WD40, and WRKY have been identified and systematically studied [32,33,34]. In this study, correlation analysis of transcription factor genes, structural genes, and differential anthocyanins led to the identification of *PmMYB5*, *PmMYB9*, *PmMYB17*, *PmMYB22*, *PmMYB23*, *PmMYB27*, *PmbHLH2*, *PmbHLH4*, *PmbHLH10*, *PmbHLH12*, *PmbHLH16*, *PmbHLH20*, and *PmWRKY4*, which play crucial roles in the synthesis of anthocyanins. The specific functions of these transcription factors can be verified in subsequent experiments.

In the process of anthocyanin synthesis, the key enzymes are categorized into two groups: upstream and downstream [35]. Upstream enzymes, including CHI, F3H, and F3′H, mainly catalyze flavonoid synthesis. Downstream enzymes, such as DFR, ANS, and UFGT, focus on anthocyanin synthesis and modification [36]. ANS plays a crucial function in the transformation of leucocyanidin into anthocyanins. Moreover, ANR is a crucial enzyme involved in the proanthocyanidin-specific pathway, competing with ANS and GST for the same substrates, leucocyanidin and anthocyanidin, respectively. The UFGT enzyme can bind glucose molecules to hydroxyl (-OH) groups within anthocyanin compounds, forming more stable 3-O-glucosylated anthocyanin compounds. Furthermore, GST also has implications for anthocyanin transport and accumulation [37]. The key genes in the anthocyanin biosynthesis pathway have been specifically studied in some plant species. For instance, the *ANR* gene in green tea ‘Shuchazao’ reduces anthocyanins to epicatechin. Elevated expression of the *ANR* gene diverts colorless anthocyanins and anthocyanidins to the proanthocyanidin synthesis branch, negatively influencing anthocyanin synthesis [38,39]. A previous study reported that *UFGT* was up-regulated in red flower buds of *P. mume* ‘Fuban Tiaozhi’, contributing to the enhanced stability of anthocyanins in red petals [40]. The *GST* family member *TT19* has been shown to contribute to anthocyanin accumulation in *A. thaliana* [41]. *MdGSTF6* plays an important role in anthocyanin transport in *M. pumila*, and its silencing affects anthocyanin accumulation in apple fruits [42]. In cotton, allelic variation in the *GhTT19* promoter is the cause of cotton color change [43]. In addition, it has been demonstrated that the transient expression of *PpGST1* in peach fruit significantly promoted anthocyanin accumulation, and its gene silencing also led to a decrease in anthocyanin content [44]. Similarly, *RsGST1* expression in radish is closely related to the accumulation of anthocyanins in the radish root bark and pulp, and overexpression of *RsGST1* in *A. thaliana* can promote the accumulation of anthocyanins [45]. This study found that the *ANR* gene was expressed at higher levels in white flowers, diverting more substrates toward the proanthocyanidin biosynthesis pathways rather than the anthocyanoside biosynthesis pathway. This shift led to the white coloration of the flowers. Our findings revealed that the purple–red variety had higher levels of expression for the majority of *UFGT* genes compared with the white-flowered variety. This suggests that *UFGT* is a crucial gene contributing to the enhanced accumulation of anthocyanins in purple–red flowers. In this study, three *GST* genes were also identified that may stimulate anthocyanin synthesis in purple–red *P. mume* flowers. At the same time, this study also found that the expression of *PmMYB17*, *PmMYB22*, and *PmMYB23* transcription factor genes was significantly positively correlated with the expression of *UFGT*, *GST*, and other genes in *P. mume*. We speculated that these three transcription factors may affect the accumulation of anthocyanins in *P. mume* by regulating the expression of *UFGT* and *GST* genes. Its function still requires further experimental verification. Additionally, *F3′5′H* is a key gene in the delphinidin synthesis pathway. In *Iris germanica*, the expression of *F3′5′H* in the deep blue outer perianth is up-regulated, resulting in a significantly higher content of delphinidin than that in the nearly white inner perianth [46]. The results of this study showed that the *F3′5′H* gene was absent in *P. mume*, which may be an important reason for the lack of the delphinidin pathway in *P. mume*. Accordingly, in molecular breeding, the introduction of an exogenous *F3′5′H* gene has the potential to produce blue or purple petals in *P. mume*.

## 4. Materials and Methods

### 4.1. Plant Material

Fresh petals of two *P. mume* cultivars, ‘Wuyuyu’ (WYY) with a purple–red flower color and ‘Sanlun Yudie’ (YD) with a white flower color, were selected as experimental materials and collected from the China National Botanical Garden (39.59 N, 116.19 E). Three biological replicates were collected for each variety. The samples were rapidly frozen with liquid nitrogen and thereafter stored at −80 °C for subsequent analyses.

### 4.2. Metabolomic Analysis and Identification of DAMs

The petals of *P. mume* were placed in a freeze–drying machine and ground to powder after vacuum freeze–drying. The sample powder (50 mg) was extracted with 1200 μL methanol–water internal standard. The extract was vortexed once every 30 min for 30 s six times. The supernatant was taken after centrifugation at 12,000 rpm for 3 min, and the sample was filtered with a 0.22 μm microporous membrane and stored in an injection bottle. Ultra-performance liquid chromatography (UPLC, ExionLC™ AD) and tandem mass spectrometry (MS/MS) were used for data collection. Metabolites were identified based on the MWDB (Metware database) [47]. Analyst 1.6.3 (AB Sciex, Framingham, MA, USA) was used to process mass spectrometry data to examine sample compounds both subjectively and quantitatively, utilizing a metabolic database stored locally. Principal component analysis (PCA) was employed to examine variations in metabolite levels across different sample groups and within each group. PCA data processing was performed using the built-in statistical prcomp function of R software (www.r-project.org/). Unit variance scaling (UV) was employed as a method of normalizing metabolite data by taking the value obtained by dividing the standard deviation of column variables after data centerization [48]. Differential metabolite screening was conducted using an FC of ≥2 or FC of ≤0.5, a *p*-value of <0.05, and VIP ≥ 1.

### 4.3. Transcriptome Analyses and Identification of DEGs

Six samples of two *P. mume* varieties were sequenced and assembled in transcriptome analyses and matched with the reference genome of *P. mume*. The Q30 value and GC content of these data were counted to analyze the quality of these sequencing data. The RNAprep Pure Polyphenol Plant RNA Kit centrifugal column) was used to extract RNA from the petals of two cultivars of *P. mume*. Ribosomal RNA was removed from total RNA to obtain mRNA. Subsequently, the RNA was broken into short fragments by adding a fragmentation buffer, and the short fragment RNA was used as a template to synthesize the first-strand cDNA with six-base random primers (random hexamers). Then, the buffer, dNTPs (dTTP, dATP, dGTP, and dCTP), and DNA polymerase I were added to synthesize the second-strand cDNA, and then the double-strand cDNA was purified using AMPure XP beads (Beckman Coulter, Beverly, MA, USA). The purified double-stranded cDNA was subjected to end repair, A-tail, and ligation of the sequencing adaptor, and then AMPure XP beads were used to select the fragment size. Finally, PCR enrichment was performed to obtain the final cDNA library. Fastp (v0.23.2) was used to control data quality during library creation and sequencing analysis [49]. The parameter setting of FASTP was as follows: length required 150; qualified quality phred 20; unqualified percent limit 50; n base limit 15. We used an Illumina novaseq 6000 for sequencing, and the sequencing strategy was PE150. A NEBNext^®^ Ultra TM II RNA Library Prep Kit for Illumina^®^ (NEB # E7775L) library construction kit was used to perform transcriptome sequencing and assembly of samples. The verification of data quality was conducted through analysis of the dispersion of the sample GC content. Clean reads were matched against the reference genome (BioProject: PRJNA720973) using HISAT2 (v2.2.1) [50]. The parameter settings of HISAT2 were as follows: dta; phred33; p 5. To obtain functional insights into the *P. mume* Unigene sequences, the nucleic acid or protein sequences were analyzed against multiple public databases such as KEGG, SwissProtein, NR, KOG, Tremble, GO, and Pfam. The DESeq2 program (v1.22.1) was employed to perform differential expression analysis between the sets of samples. The Benjamini–Hochberg method was employed to correct for the FDR in the context of multiple hypothesis testing using the probability (*p*-value) [51,52]. The DEGs were selected using certain criteria: an absolute value of |log_2_(FC)| greater than or equal to 1 and a FDR less than 0.05.

### 4.4. RT-qPCR Validation of Crucial Genes

Based on the FPKM values of each gene derived from transcriptome sequencing, 11 DEGs were selected as target genes for RT-qPCR analysis, using *PmPP2A* as the internal reference gene. The cDNA of YD and WYY petals at the initial opening stage was utilized as a template, and primer design was conducted using the Oligo (v7) software (primer sequences are provided in Appendix A). Primer specificity was confirmed using 1% agarose gel electrophoresis, and primers exhibiting a single band were selected for subsequent RT-qPCR validation. Reactions were performed using a QuantStudio 5 Real-Time PCR instrument following the instructions of the 2 × SYBR Green qPCR Master Mix kit (US EVERBRIGHT, Suzhou, China). The reaction system was as follows: 2 × SYBR Green Master Mix 5 μL, primer-F 0.5 μL, primer-R 0.5 μL, cDNA 1 μL, 10 × ROX 0.2 μL, and ddH_2_O 2.8 μL. The reaction procedure was as follows: 95 °C for 120 s, followed by 40 cycles (95 °C for 5 s and 60 °C for 30 s). The relative gene expression level was calculated using the 2^−∆∆CT^ method.

### 4.5. Integrative Analysis of DAMs and DEGs

Pearson’s correlation coefficients were calculated using SPSS 25 to examine associations between transcription factors, genes, and metabolites. DEGs and DAMs with a correlation of more than 0.99 and a *p*-value less than 0.01 were screened. A correlation network diagram was drawn using Cytoscape software (v3.9.1) [53].

## 5. Conclusions

In summary, we elucidated the differences in the types and contents of anthocyanins in purple–red and white *P. mume* through the correlation analysis of metabolomics and transcriptomics and preliminarily analyzed the gene regulatory network. Our findings indicated that 14 anthocyanins emerged as the predominant pigments influencing purple–red *P. mume* coloration. Their metabolic pathways mainly encompass two distinct branches: cyanidin and pelargonidin. Moreover, 18 key genes were identified in the anthocyanin production pathway. The gene regulatory network for *P. mume* anthocyanin synthesis was successfully constructed. These findings improve our understanding of the molecular mechanisms responsible for flower color in *P. mume*, and metabolic pathway analyses provide molecular breeding design strategies for new flower colors in *P. mume*.

## Figures and Tables

**Figure 1 plants-13-01077-f001:**
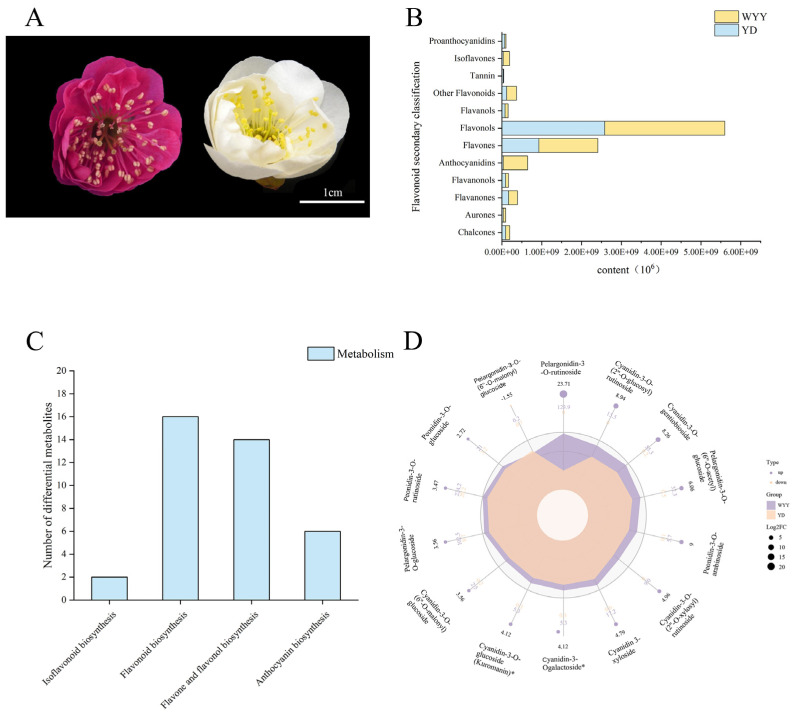
Identification of DAMs: (**A**) morphology of *P. mume*, featuring the WYY variety with purple–red flowers and the YD variety with white flowers; (**B**) differences in flavonoid metabolites between YD and WYY; (**C**) KEGG enrichment analysis of DAMs; and (**D**) anthocyanin content in petals of different *P. mume* cultivars (in units of 10^6^). * represents for isomer.

**Figure 2 plants-13-01077-f002:**
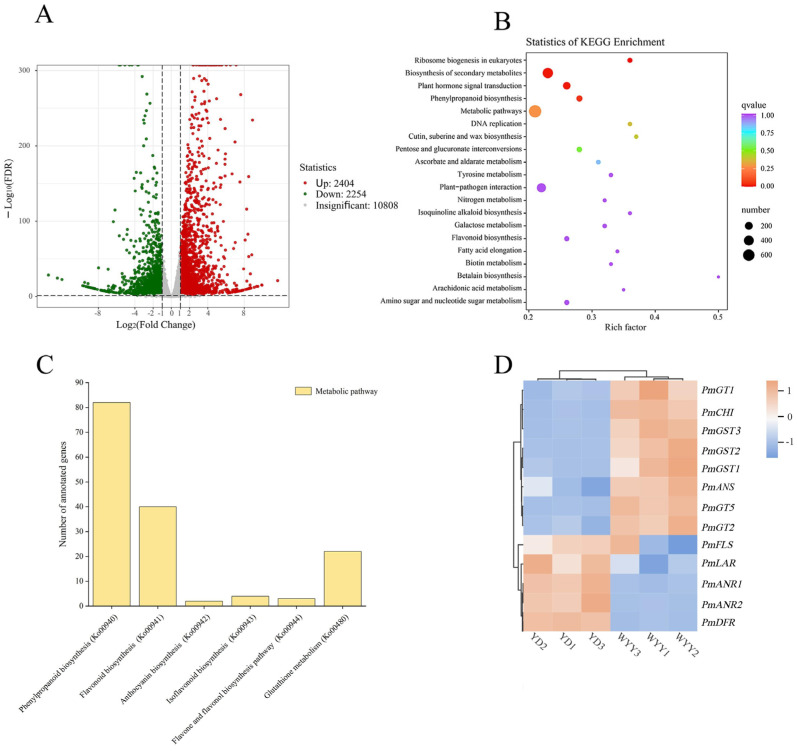
Analysis of DEGs: (**A**) number of DEGs in two *P. mume* cultivars; (**B**) top 20 KEGG-enriched DEGs represented by a bubble plot; (**C**) quantity of enriched genes in the KEGG pathway of the flavonoid metabolism pathway in *P. mume*; and (**D**) clustering heat maps illustrating the DEG expression in the anthocyanin synthesis pathway.

**Figure 3 plants-13-01077-f003:**
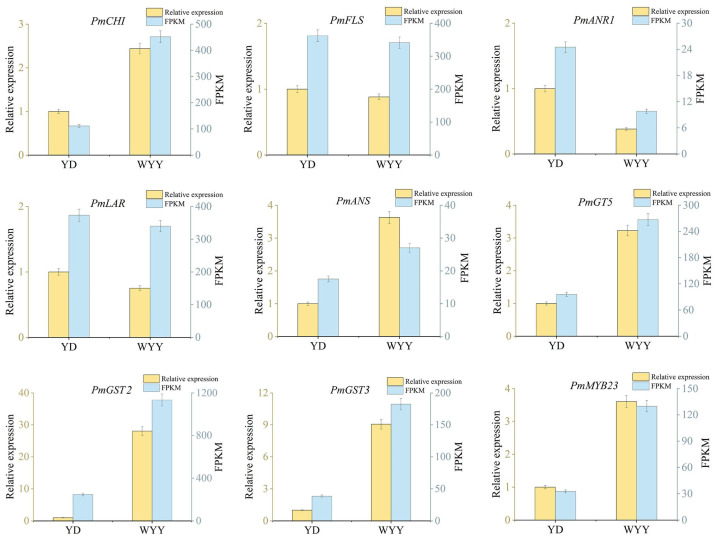
RT-qPCR and FPKM of nine DEGs identified by RNA sequencing. The *y*-axis on the left represents the relative gene expression levels (2^−∆∆Ct^) analyzed by RT-qPCR; the *y*-axis on the right shows the FPKM value obtained by RNA-seq.

**Figure 4 plants-13-01077-f004:**
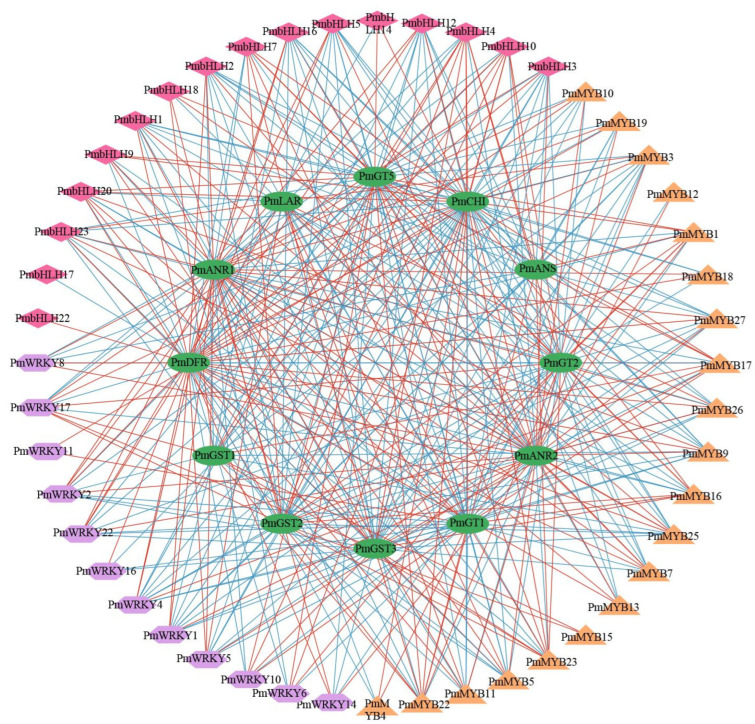
Correlation network between transcription factors and key structural genes (coefficients > 0.9 and *p* < 0.01). The inner circle represents key structural genes, and the outer circle represents the MYB, bHLH, and WRKY transcription factors. Red lines indicate positive correlations, and blue lines indicate negative correlations.

**Figure 5 plants-13-01077-f005:**
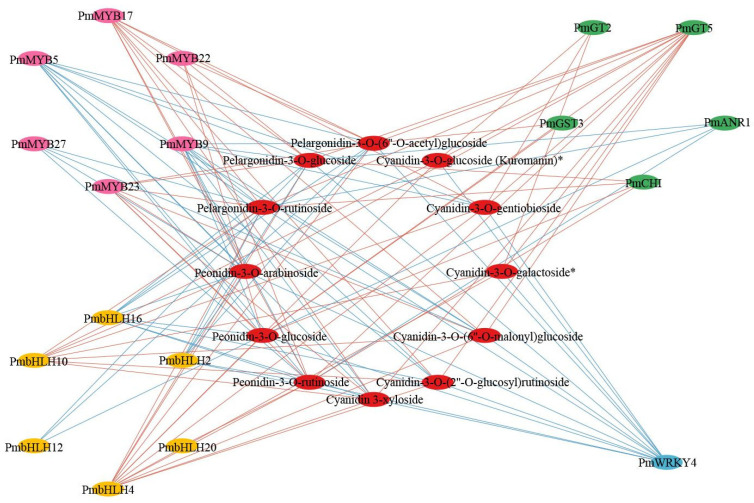
Correlation network map between DEGs and DAMs (coefficients > 0.99 and *p* < 0.01). Red lines indicate positive correlations, and blue lines indicate negative correlations. * represents for isomer.

**Figure 6 plants-13-01077-f006:**
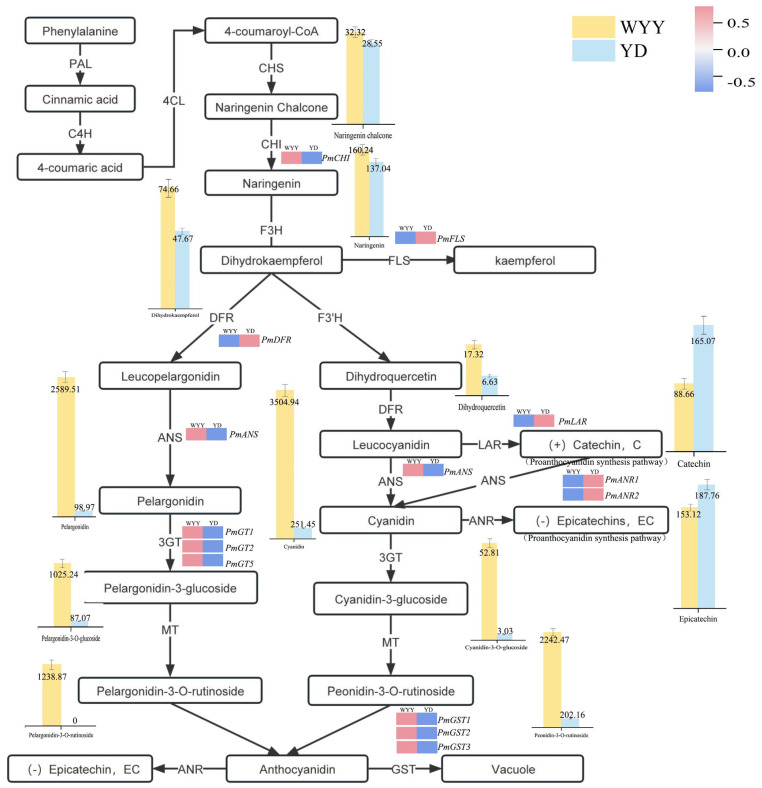
Anthocyanin biosynthesis pathways in *P. mume* flower petals. The histogram shows the contents of key metabolites in YD and WYY, and the heatmap depicts the differences in key gene expression.

## Data Availability

The datasets presented in this study can be found in online repositories. The raw reads were submitted to NCBI SRA (Sequence Read Archive, http://www.ncbi.nlm.nih.gov/sra/ (accessed on 2 January 2024)) under the accession numbers PRJNA1059508.

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
