# Peer review of "Integration of Metabolomic and Transcriptomic Analyses Reveals the Molecular Mechanisms of Flower Color Formation in Prunus mume"

_plants, 2024, doi:10.3390/plants13081077_

Round 1

Reviewer 1 Report

Comments and Suggestions for Authors

In their manuscript Wang et al. have attempted to elucidate the genes underlying differences in flower colour in two different cultivars of Prunus mume. Further, they have used metabolic profiling for detecting differences in metabolite composition between the two cultivars.

By integrating the transcriptomic and metabolomic analyses the authors claim that they could identify several genes which were positively correlated with anthocyanoside accumulation.

Intoduction

The introduction reads well. The background of the research is explained sufficiently, however, I wonder if examples could be added where the combined analysis of transcriptomic and metabolomic data helped in the detection of genes important for a certain phenotype.

Materials and Methods

line 363 to 367: Each variety was comprised of three biological replicates – should be „Three biological replicates were collected for each variety“ – what material was samples? Flowers?

Line 369 to 377: There are far too little details on the methods. How does that standard methanol extraction work? What kind of spectrometer? What kind of database? PCA was calculated with which software? What does „UV was used to process metabolite content data“ mean? There needs to be much more information on the differential metabolite screening!

Line 378 to 389: Where did the RNA come from? Which tissue? Extracted with what? How was the cDNA generated?

„Illumina provided raw sequencing data“ - ??? Sequencing data were generated on an Illumina sequencing platform (type of sequencer, sequencing company, insert size library, read length, paired or single end???)

I think there must be large parts of the methods section missing here or unclear e.g.trimmed reads were mapped against the reference genome – which genome, database number or was there any quality control of the mapping?

Line 390 to 399: No qPCR protocol given (PCR mix, PCR conditions), no manufacturer of the qPCR kit (if it was any)

Line 400 to 404: Does Cytoscape have a citation or a production company that can be added? And there need to be many more details on the network that was generated

Results

Line 133 to 150: Actually here is a lot of the information required in the materials and methods section. Why is this not described in materials and methods?

Discussion/Conclusion

The discussion requires some language editing (minor things). Overall this section of the manuscript is a bit difficult to read as the authors chose to give a lot of information on studies concerning orthologues of certain enzymes and only at the end of each section they state what their finding was concerning that enzyme in their won study. This is rather difficult to follow and should be structured differently.

The conclusion goes much beyonf wht actually has been done in the paper! The authors have shown that a number of genes are differentially expressed and that a number of metabolites accumulate to different amounts in the two cultivars investigated. However, this does not mean that they have „elucidated the detailes metabolic pathways and gene regulatory networks of anthocyanin biosynthesis“! Figure 6 provides a nice insight into possible regulatory steps in the anthocyanin pathway which may differ between the two cultivars. This can certainly be a starting point for molecular breeding once it has been shown what effects the differentially expressed genes actually do have.

Figures

Figure 3 – You need to state what the expression is relative to

Figure 6 – this is a really nice summary

Language

The Materials and Methods section needs strong improvement!

Comments on the Quality of English Language

It appears as if the manuscript was written by several different people. The introduction reads well and so does the results section. The discussion has some minor grammatical errors. And the materials and methods section is almost incomprehensible.

Author Response

Reviewer #1: In their manuscript Wang et al. have attempted to elucidate the genes underlying differences in flower colour in two different cultivars of Prunus mume. Further, they have used metabolic profiling for detecting differences in metabolite composition between the two cultivars.

By integrating the transcriptomic and metabolomic analyses the authors claim that they could identify several genes which were positively correlated with anthocyanoside accumulation.

1.The introduction reads well. The background of the research is explained sufficiently, however, I wonder if examples could be added where the combined analysis of transcriptomic and metabolomic data helped in the detection of genes important for a certain phenotype.

Response: Thank you very much. We have introduced relevant literature in the third section of the introduction, lines 84-89 to illustrate that the association analysis of transcriptome and metabolomics data helps analyze the gene expression that affects flower color.

  1. 2.line 363 to 367: Each variety was comprised of three biological replicates – should be „Three biological replicates were collected for each variety“– what material was samples? Flowers?

Response: Thank you very much. We have modified ‘Each variety was comprised of three biological replicates’ to ‘Three biological replicates were collected for each variety’. The samples are fresh petals of two P. mume varieties, which are described in line 426 of the manuscript.

  1. 3.Line 369 to 377: There are far too little details on the methods. How does that standard methanol extraction work? What kind of spectrometer? What kind of database? PCA was calculated with which software? What does “UV was used to process metabolite content data”mean? There needs to be much more information on the differential metabolite screening! 

Response: Thank you very much. The standard methanol extraction method was as follows: The petals of Prunus mume were placed in a freeze-dryer and ground to powder after vacuum freeze-drying. 50 mg of sample powder was weighed and added to 1200 μL-20°C methanol water internal standard extract. The extract was vortexed once every 30 minutes for 30 seconds for a total of 6 vortexes. The supernatant was taken after centrifugation at 12,000 rpm for 3 minutes, and the sample was filtered with a 0.22μm microporous membrane and stored in an injection bottle. Ultra Performance Chromatography (UPLC) and Tandem mass spectrometry (MS / MS) were used for data collection. Metabolites were identified based on the metware database (MWDB). PCA data processing was performed using the built-in statistical prcomp function of R software. UV (unit variance scaling) is a method for normalizing metabolite data, which is a value obtained by dividing the standard deviation of column variables after data centerization. The specific content has been supplemented in lines 435-444 of the manuscript.

  1. 4.Line 378 to 389: Where did the RNA come from? Which tissue? Extracted with what? How was the cDNA generated?

Response: Thank you very much. Total RNA was extracted from the petals of two cultivars of P. mume using RNAprep Pure polyphenol plant RNA kit (centrifugal column type). Ribosomal RNA is removed from total RNA to obtain mRNA. Subsequently, the RNA was broken into short fragments by adding fragmentation buffer, and the short fragment RNA was used as a template to synthesize the first-strand cDNA with six-base random primers (random hexamers). Then, the buffer, dNTPs (dTTP, dATP, dGTP and dCTP) and DNA polymerase I were added to synthesize the second-strand cDNA, and then the double-strand cDNA was purified by AMPure XP beads. The purified double-stranded cDNA was subjected to end repair, A-tail and ligation of the sequencing adaptor, and then AMPure XP beads were used to select the fragment size. Finally, PCR enrichment was performed to obtain the final cDNA library. We added it in lines 455-468 of the manuscript.

  1. 5.Illumina provided raw sequencing data“ - ??? Sequencing data were generated on an Illumina sequencing platform (type of sequencer, sequencing company, insert size library, read length, paired or single end???) "

Response: Thank you very much.  Transcriptome sequencing and assembly were performed using NEBNext ® Ultra TM II RNA Library Prep Kit for Illumina ® (NEB # E7775L) library construction kit samples. Sequencing was performed using Illumina novaseq 6000, sequencing strategy PE150, and sequencing company was Wuhan Matware Metabolic Co., Ltd (https://www.metware.cn/). We supplement it in lines 471-474 of the manuscript.

  1. I think there must be large parts of the methods section missing here or unclear e.g.trimmed reads were mapped against the reference genome – which genome, database number or was there any quality control of the mapping

Response: Thank you very much. The genome we refer to is the Mei flower resequencing genome led by Professor Zhang Qixiang in 2018. The BioProject number in the NCBI database is PRJNA720973. We supplemented it in line 476 of the manuscript.

  1. 7.Line 390 to 399: No qPCR protocol given (PCR mix, PCR conditions), no manufacturer of the qPCR kit (if it was any)

Response: Thank you very much. The manufacturer of the qPCR kit is US EVERBRIGHT. The reaction system was: 2 × SYBR Green Master Mix 5μ, primer-F 0.5μ, primer-R 0.5μ, cDNA 1μ, 10 × ROX 0.2μ, ddH2O 2.8μ. The reaction procedure was as follows: 95 °C 120 s, 1 cycle; 95 °C 5 s, 60 °C 30 s, cycle 40 times. We supplemented it in lines 496-499 of the manuscript.

  1. 8.Line 400 to 404: Does Cytoscape have a citation or a production company that can be added? And there need to be many more details on the network that was generated.

Response: Thank you very much. Cytoscape references have been added to line 504 of the manuscript. SPSS 25 was used to analyze the correlation between transcriptomics and metabolomics data, and the obtained correlation values were imported into Cytoscape to obtain a correlation network diagram.

  1. 9.Line 133 to 150: Actually here is a lot of the information required in the materials and methods section. Why is this not described in materials and methods?

Response: Thank you very much. We have added 130-150 lines to the materials and methods section, in lines 455-457 of the manuscript, the specific content is “Six samples of two P. mume varieties were sequenced and assembled in transcriptome analysis and matched with the reference genome of P. mume. The Q30 value and GC content of the data were counted to analyze the quality of the sequencing data.”

  1. 1The discussion requires some language editing (minor things). Overall this section of the manuscript is a bit difficult to read as the authors chose to give a lot of information on studies concerning orthologues of certain enzymes and only at the end of each section they state what their finding was concerning that enzyme in their won study. This is rather difficult to follow and should be structured differently.

Response: Thank you very much. We changed the structure of the fourth paragraph of the discussion, first briefly introduced the key enzymes in the process of anthocyanin synthesis, then introduced the research progress of related enzymes in other species, and finally introduced our findings. At the same time, we also supplemented the explanation of the relationship between transcription factors and structural genes.

  1. 1The conclusion goes much beyonf wht actually has been done in the paper! The authors have shown that a number of genes are differentially expressed and that a number of metabolites accumulate to different amounts in the two cultivars investigated. However, this does not mean that they have „elucidated the detailes metabolic pathways and gene regulatory networks of anthocyanin biosynthesis“! Figure 6 provides a nice insight into possible regulatory steps in the anthocyanin pathway which may differ between the two cultivars. This can certainly be a starting point for molecular breeding once it has been shown what effects the differentially expressed genes actuallydo have.

Response: Thank you very much. We modified the conclusion part and modified ‘elucidated the detailes metabolic pathways and gene regulatory networks of anthocyanin biosynthesis’ to ‘the differences in the types and contents of anthocyanins in purple-red and white P. mume through the correlation analysis of metabolomics and transcriptomics, and preliminarily analyzed its gene regulatory network.’ We will later identify the key genes selected to prove the actual role of these genes and strive to provide genetic resources for molecular breeding of P. mume and create new varieties of P. mume.

  1. 1Figure 3 – You need to state what the expression is relative to

Response: Thank you very much. The expression of gene is based on the expression level in the white flower variety ' YD ', which we have supplemented in lines 197-201 of the manuscript.

  1. 1Figure 6 – this is a really nice summary

Response: Thank you for your encouragement. We will continue to validate the actual functions of key genes based on the results summarized in this graph and delve deeper into the molecular mechanisms underlying their regulation.

  1. 1The Materials and Methods section needs strong improvement!

Response: Thank you very much. We supplement the materials and methods in line 435-499 of the manuscript, introduce our experimental steps in more detail, and describe in detail the standard methanol extraction, PCA calculation software, RNA extraction method and tissue, cDNA generation method, reference genome number, qPCR reaction system and so on. In addition, we have improved the language part.

  1. 1It appears as if the manuscript was written by several different people. The introduction reads well and so does the results section. The discussion has some minor grammatical errors. And the materials and methods section is almost incomprehensible.

Response: Thank you very much. Thank you for your praise for our introduction and results, and we will continue to work hard. We supplement the contents of the materials and methods so that readers can understand the contents of the manuscript more clearly. at the same time, we embellish the article in professional English. the language problems in the discussion part and the material and method part are improved.

Reviewer 2 Report

Comments and Suggestions for Authors

The manuscript is well written and it presents relevant information about metabolomics and transcriptomics regarding the molecular mechanisms of flower color in Prunus mume, furthermore, the results are of interest for molecular plant breeding not just for the genera but for other crops. It presents comprehensive relationships of transcription factors and genes that produce the red and purple colors. This information is valuable for molecular breeding. There is only a couple of finger mistakes that are not worthy to send back the manuscript for a minor revision, if the editors or the authors can correct these mistakes in the proof manuscript is approved.

The mistakes are in the sentence “In 2018, genomic data was collected by resequencing 348 P. mume species [7].” Page 2 lines 59 and 60, specifically, the word “species” is not correct, the appropriate word is “accessions”.

In page 10, in the sentence “For instance, dentification of differential metabolites…” lines 265 and 266, the correct word is “identification” not “dentification

Author Response

  1. 1.The mistakes are in the sentence “In 2018, genomic data was collected by resequencing 348 P. mume species [7].” Page 2 lines 59 and 60, specifically, the word “species” is not correct, the appropriate word is “accessions”.

Response: Thank you very much. We have corrected the word problem in line 67 of the manuscript.

  1. 2.In page 10, in the sentence “For instance, dentification of differential metabolites…” lines 265 and 266, the correct word is “identification” not “dentification”

Response: Thank you very much. We have corrected the word problem in line 284 of the manuscript.

Reviewer 3 Report

Comments and Suggestions for Authors

Review of Plants-2897425

Integration of Metabolomic and Transcriptomic Analyses Reveals the Molecular Mechanisms of Flowers Color Formation in Prunus mume

The manuscript describes experiments to enumerate the expression of genes and to quantitate metabolites involved  in flower color in P. mume.  The manuscript presents expression and metabolomic analysis of the transcriptome and metabolome of two strains P. mume that differ in flower color (white vs red).

Manuscript needs to state the common names of Prunus mume. Manuscript should explain the economic or cultural importance of this plant.

Is there a big market for the plant?  If so, how big of a market. 

Is the plant used in cultural activities?

How is P. mume propagated?  Is it clonal or outcrossing?

Are there known transformation or tissue cultures procedures for the plant?

Is the plant haploid, diploid or polyploid?

How would this impact the utility of the results?  If you can’t breed it or transform it, these are just observations that have no real utility.

Why is the “white” plant white?  Is it due to transposable element insertion?  Is it due to a spontaneous mutation?  Is it due to structural rearrangement relative to the other plant?  The sequences exist according to the background section (Resequenced 348 P. mume individuals not species).

What strain was the “Reference Genome”?  Where can the sequence of that strain be found? 

Where can the expression counts for each gene in both strains tested be found.  They should be in a supplemental table or a URL to a file of the results provided.

What options were used for FASTP?  What options were used for HISAT2?  You must state which version of all programs used along with a list of all the parameters used.

What is Analyst 1.6.3?  Who makes it?

How was the KEGG, SwissProtein, NR, KOG, Tremble, GO and Pfam database searches performed?  Was it done at the respective database websites or was it performed locally?

How was the GO enrichment analysis performed?

How was Supplemental Figure S6 created?

How was Supplemental Figure S7 created?

There must be informative legends on all the supplemental figures.

Supplemental Figure S4 presents hundreds of “DEGs” yet supplemental Table4 only lists 86 DEGs.  Is Table 4 wrong or is the title of supplemental Figure S4 wrong?

Author Response

  1. 1.Manuscript needs to state the common names of Prunus mume.Manuscript should explain the economic or cultural importance of this plant.

Response: Thank you very much. the common names of Prunus mume is Mei or Mei Flower. It has a cultivation history of more than 3,000 years in China. It represents a strong, noble, and modest character in Chinese traditional culture. P. mume have high ornamental value, which can be planted in courtyards or potted plants, and can also be used as pile scenery. This part has been supplemented in lines 59-65 of the introduction.

  1. 2.Is there a big market for the plant?  If so, how big of a market.

Response: Thank you very much. First, P. mume is a very popular and famous flower, especially in East Asia, with a profound cultural heritage that attracts many consumers. Secondly, P. mume has wide adaptability, strong resistance, and low requirements for the environment. And P. mume has a variety of colors and a long flowering period. affected by climate and varieties, P. mume can blossom from December to April to make up for the blank of winter and spring flowers, which has a high ornamental value. At the same time, the fragrant components of P. mume flowers can be extracted into perfume or used as food additives to increase flavor. P. mume fruit can also do all kinds of plum, plum sauce, and so on, with edible value. In addition, the fruit of P. mume also has medicinal value, which can promote blood circulation and detoxification. Especially in recent years, there have been continuous breakthroughs in the research on the color, resistance and fragrance of P. mume, and there will be more diverse P. mume varieties that can be applied in the future.

  1. 3.Is the plant used in cultural activities?

Response: Thank you very much. P. mume is often used in cultural activities. It represents the quality of tenacity, nobility, perseverance, and tenacious struggle in Chinese culture. It is known as the "gentleman in the flower" and is often used to describe people who move forward bravely and do not drift with the tide. It has been a fashion to watch P. mume since ancient China. Since ancient times, there are many verses, folktales and legends praising P. mume.

  1. 4.How is mumepropagated?  Is it clonal or outcrossing?

Response: Thank you very much. P. mume. are often propagated by cutting, grafting, and sowing.

  1. 5.Are there known transformation or tissue cultures procedures for the plant?

Response: Thank you very much. The tissue culture system of P. mume has been established, and its genetic transformation system is being developed.

  1. 6.Is the plant haploid, diploid or polyploid?

Response: Thank you very much. P. mume is a diploid plant.

  1. 7.How would this impact the utility of the results? If you can’t breed it or transform it, these are just observations that have no real utility.

Response: Thank you very much. Through transcriptome and metabolome analysis, we identified the difference in flower color between purple and white P. mume cultivars, and preliminarily analyzed the key genes affecting their flower color. P. mume is a perennial woody plant with a long growth cycle, so improving flower color at the genetic level can improve the efficiency of breeding. These results lay the foundation for us to further identify the functions of key regulatory genes for flower color. In the future, we can achieve molecular design and breeding of flower color by editing these key genes.

  1. Why is the “white” plant white?  Is it due to transposable element insertion?  Is it due to a spontaneous mutation?  Is it due to structural rearrangement relative to the other plant?  The sequences exist according to the background section (Resequenced 348 P. mume individuals not species).

Response: Thank you very much. The wild varieties of P. mume are predominantly white in color. Currently, there are many varieties that exhibit a white color phenotype, while also possessing other phenotypic characteristics such as flower shape, plant form, and fragrance, among others. The plant materials used in this study were selected based on their close genetic relationship and significant phenotypic differences, as determined through genomic resequencing data. It has been reported that the diverse phenotypic characteristics of P. mume color are the result of long-term evolution and artificial selection. However, it is still unclear which specific genetic variations, such as gene mutations, chromosomal recombination, and others, contribute to the color variations in P. mume. In this study, we primarily investigated the molecular mechanisms of P. mume color at the transcriptional and metabolic levels, and we have made some novel discoveries. In response to the valuable comments raised by the reviewers, we have made additional supplements and discussions in the discussion section of the manuscript.

  1. What strain was the “Reference Genome”?  Where can the sequence of that strain be found?

Response: Thank you very much. The genome is the sum of all the genetic materials of the organism. In 2018, Professor Zhang Qixiang led the re-sequencing, assembly and splicing of the P. mume, genome. Transcriptome sequencing data are randomly interrupted by mRNA. To determine which genes are transcribed from these fragments, this study compared the clean reads after sequencing quality control to the reference genome to determine the specific name and sequence of the gene. The alignment genome can be found in the NCBI database, BioProject number PRJNA720973. We have added the BioProject number of the reference genome to line 476 of the manuscript.

  1. Where can the expression counts for each gene in both strains tested be found. They should be in a supplemental table or a URL to a file of the results provided.

Response: Thank you very much. We added the expression of specific genes in the two varieties to Supplemental Table 5.

  1. What options were used for FASTP?  What options were used for HISAT2?  You must state which version of all programs used along with a list of all the parameters used.

Response: Thank you very much. Our FASTP parameter setting is: -length _ required 150 -qualified _ quality _ phred 20 -unqualified _ percent _ limit 50 -n _ base _ limit 15 (97048c8ef316). The HISAT2 parameter is set to: -dta-phred33-p 5 (49fc02ec076e). We supplemented it in lines 469-478 of the manuscript.

  1. What is Analyst 1.6.3?  Who makes it?

Response: Thank you very much. Analyst 1.6.3 is a single LC-MS / MS software for mass spectrometry data processing and analysis. The manufacturer is (AB Sciex, Framingham, MA, USA), which we have supplemented in line 444 of the manuscript.

  1. How was the KEGG, SwissProtein, NR, KOG, Tremble, GO and Pfam database searches performed?  Was it done at the respective database websites or was it performed locally?

Response: Thank you very much. KEGG, SwissProt, NR, KOG and Tremble are downloaded from the corresponding database, and the annotation results are compared with diamond blast on the local cluster. The GO comparison results are obtained by merging the annotation results of swissprot and tremble. Pfam uses hmmscan for annotation.

  1. How was the GO enrichment analysis performed?

Response: Thank you very much. GO enrichment analysis was performed using the clusterProfiler package (v4.6.0) of R software. The p-value was calculated by the hypergeometric distribution method, and the enriched p-value was corrected using the bonferroni method.

  1. How was Supplemental Figure S6 created?

Response: Thank you very much. Figure S6 is a directed acyclic graph. Enrichment analysis was performed on the differential genes between the samples, and the enriched Term was used to make a topGO directed acyclic graph. The GO directed acyclic graph is drawn using the plotGOgraph function of the R-package clusterProfiler.

  1. How was Supplemental Figure S7 created?

Response: Thank you very much. After annotating the differentially expressed genes to the KEGG database, we counted the number of differentially expressed genes contained in each KEGG pathway, and plotted a histogram based on the number of genes and the name of the KEGG pathway. The abscissa represents the number of differentially expressed genes annotated to the pathway. The ordinate represents the name of the KEGG pathway, and the rightmost label represents the classification of the KEGG pathway.

  1. There must be informative legends on all the supplemental figures.

Response: Thank you very much. We have added legend information to the relevant supplemental diagram.

  1. Supplemental Figure S4 presents hundreds of “DEGs” yet supplemental Table4 only lists 86 DEGs.  Is Table 4 wrong or is the title of supplemental Figure S4 wrong?

Response: Thank you very much. The supplementary figure S4 is a cluster heat map of all differentially expressed genes, and the expression levels of these genes have been supplemented in the supplemental table 5. Supplemental table 4 is a preliminary screening of differentially expressed genes that may be related to anthocyanin synthesis based on transcriptome metabolome association analysis. We renamed the genes according to their position on the chromosome, and supplemental table 4 is a correspondence of the ID and name of these genes.

Round 2

Reviewer 1 Report

Comments and Suggestions for Authors

Please check that all references occur in the text, I have spotted at least to instances where your citation program put an error message into your text instead of a citation.

Comments on the Quality of English Language

Language has been improved.

Author Response

Please check that all references occur in the text, I have spotted at least to instances where your citation program put an error message into your text instead of a citation.

Response: Thank you very much. We are very sorry that our carelessness led to the error of the quotation program in the manuscript. We're very sorry that our carelessness led to a citation error in the manuscript. We've checked and re-cited lines 95, 98, and 487 of the manuscript.

Reviewer 3 Report

Comments and Suggestions for Authors

Review of Plants2897425.v2

The prose of the manuscript has been improved.  The methods section has undergone extensive augmentation which was needed.

The results section still requires more revision.  It is missing key tables that would provide the data that is really not retrievable from the dense figures.

Line 93 still needs a citation

Line 97 still needs a citation

Lines 138-142  Figure S2 was unreadable.  You cannot make out the names of the metabolites.  You need a supplemental table or table listing all of the metabolites and those metabolites need to be grouped according to the text. Since you reference relative amounts in your text, this needs to be included in the table/sheet.  This is a substantial part of your data and needs to be readable. It would be best if it was in an Excel sheet.

Lines 192-193 You need to explain the significance of the gene expression density distribution Figure S3.  It is not obvious why it was included.

Lines 227-228 Exactly how did you “identify” 13 genes responsible for anthocyanin synthesis?  Did you clone the genes and rescue a phenotype?  Did you do this bioinformatically by comparing your sequences to those of another plant, say Arabidopsis?  Did you do BLAST?  Were these gene already known?  If so, you need citations for the papers that identified them.  How did you determine that you had found those genes? You need to identify which Pm gene models represent those 13 genes.  All you give is the primers that you used for qPCR.

Lines 248-250 The expression patterns were derived from the transcriptome sequencing data so they SHOULD match the RNAseq data.  This does not “confirm the reliability” of the transcriptome data. You would need some other sort of evidence to support the reliability claim.

Line 275-277 requires citations for the involvement of transcription factors  regulating the synthesis of anthocyanins.

Figure 4 is unreadable.  You cannot follow the individual lines.  This figure needs an accompanying table that lists the which TFs interact with which structural genes.

Figure 5 also needs an accompanying table listing the correlations for the same reason.

Author Response

1.The prose of the manuscript has been improved.  The methods section has undergone extensive augmentation which was needed.

Response: Thank you very much. We'll keep trying.

2.The results section still requires more revision.  It is missing key tables that would provide the data that is really not retrievable from the dense figures.

Response: Thank you very much. We added supplementary tables S7, S8, and S1, supplementary table S7 to supplement the correlation coefficient between structural genes and transcription factors in Figure 4, supplementary table S8 to supplement the correlation coefficient between anthocyanins and genes in Figure 5, supplementary table S1 to supplement 579 metabolites and their contents in supplementary figure S2.

3. Line 93 still needs a citation

Response: Thank you very much. We re-quote it and show it in [12] on line 95 of the manuscript.

4.Line 97 still needs a citation

Response: Thank you very much. We re-quote it and show it in [13] on line 98 of the manuscript.

5.Lines 138-142 Figure S2 was unreadable.  You cannot make out the names of the metabolites.  You need a supplemental table or table listing all of the metabolites and those metabolites need to be grouped according to the text. Since you reference relative amounts in your text, this needs to be included in the table/sheet.  This is a substantial part of your data and needs to be readable. It would be best if it was in an Excel sheet.

Response: Thank you very much. We added a supplementary table S1 to supplement the 549 metabolites and their contents in FigureS2. Compounds in the table are the names of metabolites, YD and WYY are two varieties of Mei flower, -1, -2, -3 represent their three replicates, YD-1, YD-2, YD-3, WYY-1, WYY-2, WYY-3 are the contents of these metabolites in their respective groups.

6.Lines 192-193 You need to explain the significance of the gene expression density distribution Figure S3.  It is not obvious why it was included.

Response: Thank you very much. Using transcriptome data to detect gene expression has high sensitivity. Figure S3 is the expression density distribution map. The density map can show the trend of gene abundance change with expression level in the sample and can reflect the interval of gene expression concentration in the sample. We have added the content in lines 168-170 of the manuscript.

7.Exactly how did you “identify” 13 genes responsible for anthocyanin synthesis? Did you clone the genes and rescue a phenotype?Did you do this bioinformatically by comparing your sequences to those of another plant, say Arabidopsis?  Did you do BLAST? Were these gene already known?  If so, you need citations for the papers that identified them. How did you determine that you had found those genes? You need to identify which Pm gene models represent those 13 genes. All you give is the primers that you used for qPCR.

Response: Thank you very much. We cloned these 13 genes, and the primers used for cloning and the full-length CDS of the obtained genes were supplemented in Supplemental Table5. We did not BLAST these sequences with Arabidopsis because the genome of Prunus mume has been sequenced, and these genes have been annotated in the genome. We compared the cloned sequence with the P. mume genome to obtain the function of the gene. We analyzed the expression patterns of these genes according to the differences in gene expression in transcriptome data and RT-qPCR, and then screened these 13 genes. Pm is the abbreviation of the Latin name of P. mume, we named these genes. The contents that need to be modified have been supplemented in lines 196-197 of the manuscript and Supplemental Table5.

8.Lines 248-250 The expression patterns were derived from the transcriptome sequencing data so they SHOULD match the RNAseq data.  This does not “confirm the reliability” of the transcriptome data. You would need some other sort of evidence to support the reliability claim.

Response: Thank you very much. Before analyzing the transcriptome data, we carried out strict quality control of the transcriptome data, including G30 value and GC content, which can explain the reliability of the transcriptome data. These data have been explained in the 437-439 lines of the manuscript. The above results can explain the reliability of the transcriptome data. We analyzed the expression patterns of these genes by RT-qPCR, and preliminarily identified these genes to further clarify their functions. The content of the statement was revised in the 212-223 lines of the manuscript.

9.Line 275-277 requires citations for the involvement of transcription factors regulating the synthesis of anthocyanins.

Response: Thank you very much. We added references in line 236 of the manuscript, which are references of [14] and [15]. These two references discussed the role of MYB, bHLH, WD40 and WRKY transcription factors in the process of anthocyanin.

10.Figure 4 is unreadable.  You cannot follow the individual lines.  This figure needs an accompanying table that lists the which TFs interact with which structural genes.

Response: Thank you very much. We added supplementary table S7 to supplement the correlation coefficient between structural genes and transcription factors in Figure 4.

11.Figure 5 also needs an accompanying table listing the correlations for the same reason.

Response: Thank you very much. We added supplementary table S8 to supplement the correlation coefficient between anthocyanin and genes in Figure 5.

Round 3

Reviewer 3 Report

Comments and Suggestions for Authors

Re-review of plants-2897425

The Authors were responsive to my issues. The Authors also appear to have been responsive to the other reviewer’s issues and the manuscript reads much better now.

I recommend accepting the manuscript.